# Pre-Sleep Casein Protein Ingestion Does Not Impact Next-Day Appetite, Energy Intake and Metabolism in Older Individuals

**DOI:** 10.3390/nu12010090

**Published:** 2019-12-28

**Authors:** Stephen Morehen, Benoit Smeuninx, Molly Perkins, Paul Morgan, Leigh Breen

**Affiliations:** 1School of Sport, Exercise and Rehabilitation Science, University of Birmingham, Birmingham B15 2TT, UK; stephenmorehen@gmail.com (S.M.); smeuninx@hotmail.com (B.S.); P.T.Morgan@bham.ac.uk (P.M.); 2Department of Sport and health Sciences, University of Exeter, Exeter EX1 2LU, UK; mmp209@exeter.ac.uk

**Keywords:** pre-sleep protein, sarcopenia, ageing

## Abstract

Maintaining adequate daily protein intake is important to maintain muscle mass throughout the lifespan. In this regard, the overnight period has been identified as a window of opportunity to increase protein intake in the elderly. However, it is unknown whether pre-sleep protein intake affects next-morning appetite and, consequently, protein intake. Therefore, the purpose of the current study was to investigate the effects of a pre-sleep protein drink on next-morning appetite, energy intake and metabolism. Twelve older individuals (eight males, four females; age: 71.3 ± 4.2 years) took part in a single-blind randomised cross-over study. After a standardised dinner, participants consumed either a 40-g protein drink, isocaloric maltodextrin drink, or placebo water control before bedtime. Next-morning appetite, energy intake, resting metabolic rate (RMR), respiratory exchange rate (RER), and plasma acylated ghrelin, leptin, glucose, and insulin concentrations were assessed. No between-group differences were observed for appetite and energy intake at breakfast. Furthermore, RMR, RER, and assessed blood markers were not significantly different between any of the treatment groups. Pre-sleep protein intake does not affect next-morning appetite and energy intake and is therefore a viable strategy to increase daily protein intake in an older population.

## 1. Introduction

Age-related declines in skeletal muscle mass (sarcopenia) and strength (dynapenia) are positively associated with functional dependency, frailty, falls, metabolic disease, and early death [1,2,3,4,5]. Sarcopenia progression is underpinned by aspects of primary chronological ageing or secondary ageing (i.e., biological factors including nutrition and inactivity). Factors related to secondary ageing such as inactivity [6], a reduced energy/protein intake [7] and the presence of chronic low-grade inflammation [8] can be targeted and either slowed or reversed through interventions. The rapid global expansion in the number of older individuals is expected to present an unprecedented burden for national healthcare systems [9,10]. Indeed, in the UK alone, costs associated with sarcopenia are currently estimated to be in excess of £2.5 bn per annum [11]. Hence, the development and/or refinement of strategies to mitigate sarcopenia is of the utmost importance for healthy population ageing and socio-economic prosperity. 

The maintenance of skeletal muscle mass hinges on the fine balance between rates of muscle protein synthesis (MPS) and breakdown (MPB). In younger individuals, dietary amino acid provision alone or in close proximity to resistance exercise has the ability to increase MPS and suppress MPB, leading to skeletal muscle accretion over time. This anabolic response to dietary protein is, however, blunted in older compared with younger individuals, such that a greater protein dose is needed to elicit a similar muscle anabolic response. A retrospective study by Moore and colleagues revealed that maximal stimulation of MPS occurred at 0.24 g·kg body mass^−1^ and 0.40 g·kg body mass^−1^ in young and older individuals, respectively [12]. Based on these per meal protein intakes, it has been suggested that the current recommended dietary allowance (RDA) for protein of 0.8 g·kg^−1^·day^−1^ is suboptimal for the maintenance of skeletal muscle mass with advancing age. However, increasing protein intakes up to 1.2–2.0 g·kg^−1^·day^−1^ has been shown to reduce long-term lean mass loss in the elderly [13], and has been shown to have no negative side-effects on health outcomes such as kidney function [14]. Oftentimes the overnight period is overlooked as a window of opportunity to increase protein intake. Recently, it was shown that pre-sleep protein ingestion in older individuals is successfully digested, absorbed and incorporated in de novo protein synthesis during the overnight period [15]. Thus, the overnight period, which under normal circumstances is linked to postabsorptive muscle catabolism, could be used to maximise muscle anabolism with important implications in sarcopenia [16,17]. In addition to the potential for overnight muscle anabolism, pre-sleep protein ingestion has been reported to increase next morning resting metabolic rate (RMR) and fat oxidation, potentially conferring beneficial changes in body composition over time [18]. 

At present, the effect of pre-sleep protein ingestion on next morning appetite and metabolic health parameters in older individuals is unclear. Protein consumption has been suggested to induce feelings of satiety which can lead to a reduced caloric intake during subsequent meals [19,20]. However, in young and older individuals next-morning appetite, as assessed through the visual analogue scale, was not affected by pre-sleep protein ingestion [15,21]. Besides protein’s potential satiating effect, protein consumption can increase RMR which is positively associated with the size of a self-determined meal [22]. As older individuals exhibit reductions in appetite due to alterations in physiological (impaired sensory perception and chewing capability), psychosocial (loneliness), and orexigenic and anorexigenic hormonal factors (ghrelin, leptin) [23,24,25], it is imperative to understand whether ingestion of a large dose of pre-sleep protein would further impact next-morning protein/energy intake. If pre-sleep protein consumption attenuates appetite and, consequently, energy/protein intake at breakfast the next morning, it might render pre-sleep protein ingestion counterproductive, as this could possibly hinder skeletal muscle maintenance.

Therefore, the aim of the present study was to determine whether a pre-sleep casein protein beverage would influence next-morning indices of appetite, energy intake and metabolism in older individuals compared with an energy-matched, protein-free drink and a noncaloric placebo drink. We hypothesized that pre-sleep casein protein and an energy-matched carbohydrate drink would increase feelings of satiety in the immediate post-prandial period compared with a non-energetic placebo but would not adversely affect next morning indices of appetite, ad libitum energy intake, or metabolic parameters in older individuals. 

## 2. Materials and Methods 

### 2.1. Participants

Twelve healthy older participants (8 male) (age: 71.3 ± 4.2 yrs; body mass: 71.1 ± 13.4 kg; BMI: 23.7 ± 3.4) were recruited from the local Birmingham (UK) community. Participants were excluded from study participation if they performed regular structured training more than three times a week, smoked, had Type I/II diabetes, or any medical condition that could cause discomfort in response to food ingestion. Furthermore, participants were excluded if they presented an eating disorder, allergy or food intolerance to any of the ingredients present in the pre-sleep beverages, standardised dinners or ad libitum breakfast. Participants provided written informed consent and general health questionnaires before the start of any experimental trials. Ethical approval for the present study was granted by the University of Birmingham School of Sport, Exercise and Rehabilitation Science Research Ethics Committee (ERN_18-1221) and the study was completed in accordance with the Declaration of Helsinki. 

### 2.2. Study Design

Following a preliminary visit to assess participants’ study eligibility, anthropometric characteristics, physical function and dietary habits, participants underwent 3 experimental trial visits in a randomized single-blind crossover fashion. Participants were randomized using the randomization.com website, creating random and balanced permutations of treatments for each subject. The evening prior to each experimental trial visit participants were provided with a standardised dinner and pre-sleep treatment beverage. The following morning participants attended the laboratories at the School of Sport, Exercise and Rehabilitation Sciences at 07:00 to undergo RMR measurements followed by multiple blood sampling and the consumption of an ad libitum breakfast. Subjective measurements of appetite were carried out the evening prior and during the experimental trial visit. Experimental trials were identical, apart from the pre-sleep treatment beverage and separated by 7–10 days. All study visits took place at the School of Sport, Exercise and Rehabilitation Sciences of the University of Birmingham. An overview of the experimental design is provided in Figure 1.

### 2.3. Preliminary Visit

Participants met with the principal researcher prior to study enrolment to discuss the study procedures and confirm eligibility. Participants were asked to complete a general health questionnaire and food preference form. Following this, participants’ height, body mass and BMI were obtained. A short physical performance battery (SPPB) [26] was completed to ensure participants’ basic levels of physical independence and gave scores (1–4) for balance, gait speed and leg strength. To assess habitual dietary intake, participants were asked to complete a 3-day weighed food diary to be completed.

### 2.4. Experimental Visits

#### 2.4.1. Standardization of Evening Meal

A standardised dinner (50% CHO, 32% FAT, 18% PROT) was provided to be consumed the evening prior to each experimental visit. Energy provision was manipulated based upon the obtained food diaries to match the participant’s habitual energy intake during dinner. Food diaries were analysed using Dietplan (Dietplan 7, Forestfield Software Ltd., Horsham, UK) and macronutrient breakdown was determined.

#### 2.4.2. Pre-Sleep Beverages

Participants were asked to consume 1 of 3 pre-sleep treatment beverages in a randomised cross-over fashion at 22:00 the evening prior to each experimental trial visit. Beverages consisted of either water (WP), 48 g of casein protein powder (MyProtein, Northwich, UK) providing a 40 g protein-dose (CP), or 42 g of maltodextrin powder (MyProtein, Northwich, UK) isocaloric to the casein drink (MD). Participants were given a shaker to take home and instructed to mix the casein or maltodextrin powder with 400 mL of water. Non-caloric vanilla flavouring (MyProtein, Northwich, UK) was added to the treatment beverages to improve palatability. A 40-g casein protein-dose was chosen based upon previous research findings demonstrating its ability to effectively stimulate overnight MPS in older adults [15]. The nutritional content of each treatment beverage is presented in Table 1. 

#### 2.4.3. Subjective Measurement of Appetite and Sleep Quality

Feelings of appetite and thirst were assessed using a 100 mm visual analogue scale (VAS) covering 12 questions regarding hunger, fullness, desire to eat and thirst. Each question was accompanied by two opposing statements interspersed by a 100 mm horizontal line. Participants were instructed to mark the 100 mm VAS at the point most accurately representing their current feeling. The distance between the mark and beginning of the 100 mm VAS was measured to determine a score between 0 and 100. For comparison, a baseline VAS for pre-sleep appetite and sleep quality in the absence of a pre-sleep beverage was performed in the week leading up to the first experimental trial. During each experimental trial visit, an appetite VAS was completed at the following 12 consecutive time points: immediately before, after and 30 min after pre-sleep treatment beverage consumption, upon waking, upon arrival at the laboratory, post-RMR measurement, pre-breakfast, immediately post-breakfast, 30 min post-breakfast, pre-lunch, pre-dinner and finally pre-bedtime. VAS’s are shown to have high validity and reproducibility in the measurement of subjective appetite [27]. As the focus of the current investigation was hunger and desire to eat, only the first 4 questions of the VAS relating to these domains were included for analysis. VAS scores for appetite were split into two time-frames; pre-treatment (22:00) to pre-breakfast (08:50), and post-breakfast (09:50) to pre-sleep. Therefore, two baseline values were used in the statistical analyses, i.e., the pre-treatment value (22:00) and post-breakfast value (09:05). These timeframes are referred to as time-frame 1 and time-frame 2, respectively, throughout the manuscript.

Sleep quality was determined using the Leeds Sleep Evaluation Questionnaire (LSEQ) [28], consisting of 10 questions split into four domains (getting to sleep, quality of sleep, waking from sleep and behaviour following wakefulness). A 100 mm VAS was used to determine the response to each question, and an average score was calculated for each domain. Participants completed one baseline LSEQ upon waking in the week leading up to the experimental trial, and on the morning of the experimental trial. 

#### 2.4.4. Resting Metabolic Rate Measurement 

Participants rested supine for 30 min in a supine position on a bed in a quiet, temperature regulated (22–24 °C), dim lit room. Expired gasses were collected over the final 20 min of the measurement period using a mouthpiece attached to a Douglas bag. A calibrated gas analyser was used to determine the percentage of O_2_ and CO_2_ in the expired air. Volume and temperature of expired air was assessed using a dry gas meter, and barometric pressure of the laboratory determined in which gas analysis was taking place. Participants were instructed to lie as still as possible, breathe normally, remain quiet and stay awake. The Weir formula [29] was used to calculate RMR and accordingly, resting energy expenditure.

#### 2.4.5. Ad Libitum Breakfast and Assessment of Energy Intake

An ad libitum breakfast was provided to participants during each experimental visit. To determine energy intake and macronutrient composition, all breakfast items were weighed before and after consumption, and analysed using Dietplan software (Dietplan 7, Forestfield software Ltd., UK). Participants were seated in a comfortable, quiet environment and instructed to eat as much or as little as they desired. The breakfast provided consisted of the following components: water, Kellogg’s cornflakes, Sainsbury’s medium white bread, Sainsbury’s raspberry jam, Sainsbury’s semi-skimmed milk (all Sainsbury’s Stores Ltd.), Clover margarine, and yoghurt (Danone Activia). Participants were unaware that their food consumption was being measured in order to avoid any confounding factors

#### 2.4.6. Blood Sampling and Analyses

Venous blood samples were obtained from an antecubital forearm vein at each experimental trial visit immediately before, immediately post and 30 min post ad libitum breakfast consumption. Ten ml blood samples were obtained and decanted evenly between serum-separator and EDTA vacutainers (BD, Oxford, UK), centrifuged at 3000 RPM for 10 min at 4 °C, and stored at −80 °C. Plasma glucose concentrations were analysed in duplicate using an RX Daytona^+^ analyser. Plasma acylated ghrelin and plasma leptin were analysed using commercially available enzyme-linked immunosorbent assays (ghrelin: Bertin Pharma, Montigny le Bretonneux, France; leptin: Mercodia AB, Uppsala, Sweden), following the manufacturer’s instructions. Samples were batch analysed and samples from each participant were analysed on the same plate. The intra-and inter-assay coefficient of variation for acylated ghrelin and leptin were <5% and 10%, respectively. Plasma insulin concentrations were determined using a commercially available enzyme-linked immunosorbent assay according to the manufacturer’s instructions (R&D Systems Inc., Minneapolis, MN, USA). All plasma analyses results are presented for *n* = 9 as some blood samples were missing for 3 participants due to cannula issues.

### 2.5. Statistics

All statistical analyses were completed using Graphpad Prism (Version 8.3). One-way repeated measures ANOVAs were conducted to examine differences in RER, RMR, subjective ratings of sleep quality, relative energy intake at breakfast and markers of insulin resistance. Two-way repeated measures ANOVAs assessed interactions between time-point and condition for subjective hunger scores and blood plasma analyses following consumption of breakfast. Pearson’s correlational analysis was performed to determine whether pre-breakfast ratings of hunger correlated with actual energy intake at breakfast. Tukey post-hoc analysis was conducted to identify specific differences where a main effect or interaction was found. Alpha level was set to *p* < 0.05. All data are reported as mean ± standard error and error bars on figures are indicative of standard error of man. 

## 3. Results

### 3.1. Appetite and Sleep Quality

Throughout time-frames 1 and 2, no between group differences were found for all four questions. During time-frame 1, hunger (Figure 2A) was significantly elevated from baseline at 08.50 in all three groups (*p* < 0.02 for all three groups), at 08:30 in MD (*p* = 0.016) and WP (*p* = 0.002), and at 08:00 in WP (*p* = 0.002). Satisfaction (Figure 2B) was significantly decreased from baseline values at 08:30 (*p* = 0.039) and 08:50 (*p* = 0.009) in CP, at 08:00 (*p* = 0.0038) and 08:50 (*p* = 0.009) in MD, and at 08:00 (*p* < 0.001), 08:30 (*p* < 0.001) and 08:50 (*p* < 0.001) in WP. Fullness (Figure 2C) was significantly lower compared with baseline values for CP, MD, and WP at 8:00, 08:30, and 08:50 (*p* < 0.029 for all), whilst significantly increased at 22:15 (*p* = 0.029) and 22:30 (*p* = 0.048) in CP only. The desire to eat (Figure 2D) was increased from baseline at 08:00, 08:30, and 08:50 in both MD and WP (*p* < 0.02 for all), whilst only increased at 08:30 (*p* = 0.009) and 08:50 (*p* = 0.002) in CP. During time-frame 2, values at pre-lunch and pre-dinner for all three groups were significantly different from their respective baseline values at 09.05h for all questions (*p* < 0.004 for all). Satisfaction and fullness were, respectively, increased at 09.35h (*p* = 0.044) and decreased (*p* = 0.0499) from baseline values in CP only. 

Sleep quality, as assessed by the four LSEQ domains, was not affected by any of the treatment beverages (GTS: *p* = 0.963, QOS: *p* = 0.926, WFS: *p* = 0.156, BFW: *p* = 0.883). Furthermore, baseline sleep quality and sleep quality prior to any of the experimental visits did not differ, suggesting that consumption of a pre-sleep beverage did not impact sleep quality. Sleep quality data was uploaded as a Appendix A).

### 3.2. Energy Intake and Metabolic Measurements

Absolute total energy intake at breakfast was not different across the treatment groups (Figure 3A), with a similar response for relative carbohydrate, fat and protein intake (Figure 3B). No differences were observed between the relative protein, carbohydrate and fat intake of the experimental ad libitum breakfast and habitual breakfast consumption, assessed by the 3-day weighed food diaries (Appendix A). Furthermore, no significant relationship between the subjective ratings of hunger before the experimental breakfast and subsequent energy intake at breakfast were observed in any of the experimental conditions (CP: r = −0.129, *p* = 0.689; MD: r = −0.099, *p* = 0.759; WP: r = −0.119, *p* = 0.712.). No significant differences between groups were observed for RMR (*p* = 0.348) and RER (*p* = 0.874) (Figure 3C,D).

### 3.3. Glucose, Insulin, Ghrelin and Leptin Response 

No overall and time-point specific between-group differences were observed for any of the plasma analyses (Figure 4). Glucose was significantly elevated from pre to 30 min post (*p* = 0.023) and from immediately post to 30 min post (*p* = 0.04) breakfast consumption in MD only. Insulin concentrations at 30 min post breakfast consumption were significantly increased from pre (*p* < 0.001) and immediately post breakfast (*p* < 0.043) consumption values in all three groups. Plasma insulin concentrations in MD were elevated immediately post compared with pre breakfast values (*p* = 0.006). Acylated plasma ghrelin concentrations were decreased 30 min after breakfast consumption compared with baseline values in CP (*p* = 0.038). In MD acylated plasma ghrelin concentrations significantly decreased from immediately post to 30 min post breakfast consumption (*p* = 0.012), whilst acylated plasma ghrelin concentrations in WP were lower 30 min post compared with pre (*p* = 0.029) and immediately post (*p* = 0.012) breakfast values. No significant within group differences were found for plasma leptin concentrations. 

## 4. Discussion

Adequate dietary protein intake with age is important to maintain skeletal muscle mass. The present study comprehensively investigated the effects of a pre-sleep protein beverage containing 40 g of casein protein on next-morning appetite levels. In our hands, a pre-sleep protein beverage did not negatively affect next-morning appetite as determined by the consumption of an ad libitum breakfast, visual analogue scales, and plasma ghrelin, leptin, insulin and glucose concentrations. Furthermore, no differences were found in RMR and RER measurements the morning after consuming any of the pre-sleep treatment beverages. Therefore, the consumption of a pre-sleep protein beverage could be a viable strategy to maintain or increase protein intake without inferring negative effects on next-day calorie consumption.

Protein nutrition is key to maintain muscle mass and attenuate the progression towards sarcopenia [30]. The favourable changes induced by dietary protein intake in older individuals reach beyond those observed on muscle mass [7] and body composition [31] and include reductions in plasma triacylglycerols, total cholesterol, very low-density lipoproteins and hepatic lipid content [32]. This raises a clear case for protein consumption to be above the current RDA of 0.8 g·kg^−1^·day^−1^ in older individuals to maintain overall health. Recently, the overnight period has been identified as a potential time-frame to increase protein intake. Indeed, several studies in both young and older individuals have shown the ability of an overnight nasogastric or pre-sleep protein feed to induce an increased MPS response during an otherwise catabolic period [21,33,34]. Nevertheless, promoting a pre-sleep feeding strategy would be detrimental to muscle maintenance if it were to decrease next-morning appetite and consequently protein intake. The present study thoroughly investigated appetite feelings the day following the consumption of a bedtime protein beverage and found no adverse effects on next-morning appetite when compared with a water placebo or isocaloric maltodextrin drink. Our results confirm earlier suggestions from Kouw and colleagues [34] and Madzima et al. [35] showing no effects of a pre-sleep protein drink on next morning indices of appetite. Studies that have reported a satiating effect of protein on next morning appetite might have been influenced by the macronutrient composition of the pre-sleep treatment, or the timing and mode of protein delivery. In this regard, Ormsbee et al. [36] found that when chocolate milk is consumed before bedtime, next morning appetite is reduced in young females. Furthermore, Groen et al. [33] showed reductions in next-morning appetite in elderly men following intragastric administration of 40 g of casein during sleep. The presence or absence of a satiating effect inflicted by protein consumption is likely determined by the amount of time elapsed between pre-sleep protein consumption and next-morning measures of appetite. It is therefore not surprising that these feelings of fullness with protein intake typically occur during the acute postprandial period [37,38,39]. Further evidence supporting the lack of a satiating protein effect in the current study can be found in the absence of a significant between-group difference in total energy and relative macronutrient consumption during the ad libitum breakfast, similar to observations in younger individuals in the study of Lay et al. [40]. Finally, energy intake tended to be higher in response to habitual breakfast intake as opposed to the trial breakfast, which might be a consequence of the relatively limited number of foods served at the trial breakfast. To overcome this hurdle, future investigations should include a wider variety of foods to match personal dietary preferences. It is important to note that the present study utilised visual analogue scales to assess subjective feelings of hunger and appetite, which might not correlate with energy intake [41]. This is reflected in our inability to find a correlation between pre-breakfast hunger ratings and subsequent energy intake, in spite of no between-group differences between these variables. Future research should ensure actual energy intake is assessed alongside appetite for the entirety of the following day after pre-sleep protein ingestion, not just breakfast, as one is not necessarily indicative of the other.

Sleep is a key aspect in maintaining a healthy lifestyle. Not only does sleep play an important part in regulating our internal molecular mechanisms (e.g., hormone release), it also drives our circadian rhythms, and consequently whole-body metabolism [42]. It has been shown previously that a disrupted or irregular sleep pattern can lead to obesity and insulin resistance [43,44]. Factors contributing to these disrupted sleep patterns are multifactorial and include dietary habits [45]. In this regard, a study by Crispim and colleagues revealed a negative relationship between nocturnal food intake and sleep quality in healthy individuals [46]. In the present study sleep quality was assessed using the LSEQ, and was not significantly different between any of the treatment groups and from baseline (pre-intervention) measurements, suggesting no effect of the pre-sleep beverage on overall sleep quality (Appendix A). Nevertheless, multiple participants anecdotally reported a disturbed sleep after consuming one of the provided treatment beverages. The amount of liquid (400 mL) consumed before bed-time led to increased urination at night. These anecdotal reports inferring sleep disruption could be associated with a deviation from habitual dietary routines. A study by Adam [47] elegantly showed that individuals who habitually ate before bedtime showed a decreased sleep quality when consuming non-caloric capsules instead. The opposite held true too, individuals who were accustomed to abstaining from any pre-sleep food intake became more sleep deprived when drinking milk before bedtime. The authors concluded that a departure from a person’s habitual evening food intake would impair subsequent sleep. Therefore, the question remains whether these anecdotal reports of increased urination during the night, would have persisted and impacted upon overall sleep quality if a pre-sleep beverage were to be provided chronically.

Feelings of hunger are strongly dictated by the interplay of several hormones, most notably insulin, leptin and ghrelin. Insulin is secreted by the pancreatic beta cells and plays a role in glucose metabolism, leptin is secreted by adipose tissue and is a satiety factor, whilst ghrelin is primarily secreted by the stomach and duodenum and is an orexigenic ligand [48]. As amino acids are believed to suppress feelings of hunger, we measured plasma leptin, acylated ghrelin, glucose, and insulin concentrations before, after and 30 min after the ad libitum breakfast consumption, and whether these parameters would be affected by the pre-sleep treatment beverage. It is well-established that a pre-prandial increase in circulatory ghrelin concentrations is observed before meal-consumption [49]. In the present study, no between-group differences were observed in pre-breakfast concentrations of plasma acylated ghrelin, leptin, or any of the other measured blood markers. These observations strongly suggest that the pre-sleep beverage did not affect any of the hunger related hormones (pre-breakfast). It is therefore not surprising that calorie intake at breakfast was equal between treatment groups. Plasma acylated ghrelin concentrations were decreased from either pre-breakfast values in CP and WP or immediately post-breakfast values in MD. This finding is in line with other research showing a postprandial nadir in plasma ghrelin concentrations 60 min after consuming a mixed-macronutrient meal [50]. Plasma insulin concentrations were increased 30 min after breakfast consumption in the present study, irrespective of treatment group, which is supported by previous findings [51,52]. Furthermore, an inverse relationship between insulin and ghrelin levels has previously been reported. A study by Saad et al. [53] demonstrated that an insulin infusion led to suppressed ghrelin concentrations up to 15 min after cessation of the infusion. This inverse relationship between plasma insulin and acylated ghrelin levels was also observed in the present study. Plasma leptin concentrations did not change in our study, which is likely due to the relatively short time-frame over which blood samples were collected. A study by Romon and colleagues [54] showed a rise in leptin concentrations four hours after meal-consumption. Further research is required to fully elucidate the acute postprandial effects of a pre-sleep protein beverage on the regulation of appetite hormones, and how this relates to feelings of satiety. This could inform the optimal dose and timing of a pre-sleep protein beverage without affecting next-morning appetite. 

Protein and energy content of a meal have been shown to influence food-induced thermogenesis [55]. In the present study, we anticipated to see an effect of the pre-sleep protein, and possibly carbohydrate, beverage on RMR and RER. However, we were unable to detect any differences between treatment groups for the aforementioned variables. Studies investigating the effects of a pre-sleep snack on next-morning RMR measurements are equivocal, with some showing an increase [36] and others no effect [40,56,57]. Discrepancies in these findings are likely related to the elapsed time between pre-sleep snack consumption and RMR measurement, with shorter time-frames increasing the likelihood of finding an effect. Indeed, a study conducted by Ormsbee and colleagues in young female runners found an increase in next-morning RMR when 335 mL of chocolate milk (containing 12 g of protein and 30 g of CHO) was consumed 30 min before bedtime and ~7h before the RMR measurement [36]. In contrast, a study by Lay and colleagues failed to observe an increase in RMR when a pre-sleep snack was consumed ~10 h earlier in mildly overweight young men [40]. These findings were confirmed by Kinsey et al, who reported no increase in RMR ~8 h post bedtime snack consumption in obese young men [56]. The absence of any changes in next-morning RMR is not surprising as previous metabolic ward investigations have demonstrated that diet-induced thermogenesis returns to basal values within six hours of consuming a pre-sleep snack [58]. Finally, sleep quality and sleep duration can impact RMR, with shorter sleeping times leading to a decrease in RMR [59]. As sleep quality was unaffected by the treatment beverage, this likely did not impact RMR measurements. Besides RMR, next-morning substrate utilisation (RER) was assessed in the present study, and was not affected by the pre-sleep treatment beverage. The absence of any effect on RER is conceivably a consequence of unaltered insulin concentrations between groups. Substrate utilisation is strongly dictated by circulatory insulin levels, with elevated insulin concentrations shifting substrate utilisation towards carbohydrate oxidation [60]. In this regard, a study by Pennings et al. [61] has previously shown that casein ingestion resulted in a lower insulinemic response compared with whey protein and casein hydrolysate. This decreased insulinemic response could shift fuel utilization towards increased fat oxidation rates and explain the results observed by Madzima and colleagues demonstrating increased RER values when whey protein or carbohydrates were consumed pre-sleep compared with a placebo control [18]. It could hence be speculated that casein ingestion would lead to lower plasma insulin concentrations compared with the isocaloric carbohydrate drink. Nevertheless, next-morning plasma glucose and insulin concentrations were similar between treatments. This is to be expected as insulin and glucose levels return to baseline levels ~2 h after an oral glucose tolerance test [62]. Therefore, the comparable plasma insulin concentrations between treatments likely underlies the similar RER values.

## 5. Conclusions

The results of the present study suggest that 40 g of casein protein ingested as a beverage prior to sleep, does not adversely affect sleep quality or next-morning indices of appetite, energy intake or metabolism in older individuals. As such, the implementation of pre-sleep protein strategies to increase daily protein consumption in older individuals is a viable option and should not elicit reductions in energy (and subsequently protein) intake at breakfast time or other meals the following day. Future research should look to expand these findings by investigating different food forms of pre-sleep protein snack (liquid vs. solid), as well as assessing energy and protein intake, appetite, and activity throughout the entirety of the day following experimental visits to determine whether pre-sleep protein has any effect on whole-day behaviours. Furthermore, based upon the above and previous evidence of the acute efficacy of a pre-sleep protein snack on muscle anabolism, more longitudinal studies should explore whether bedtime protein consumption could attenuate the sarcopenic progression.

## Figures and Tables

**Figure 1 nutrients-12-00090-f001:**
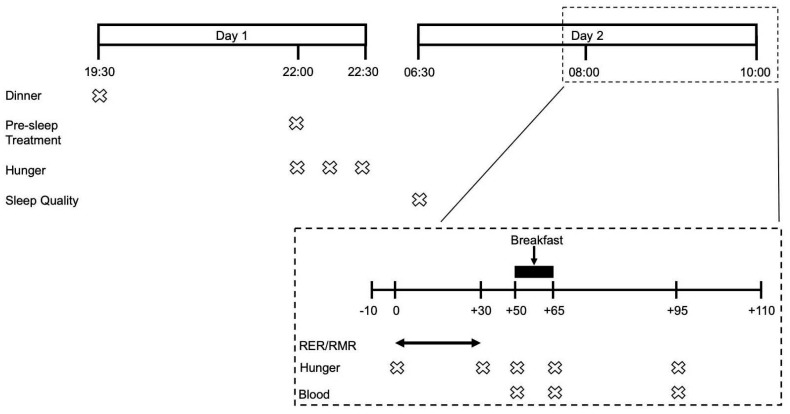
Timeline of an experimental visit.

**Figure 2 nutrients-12-00090-f002:**
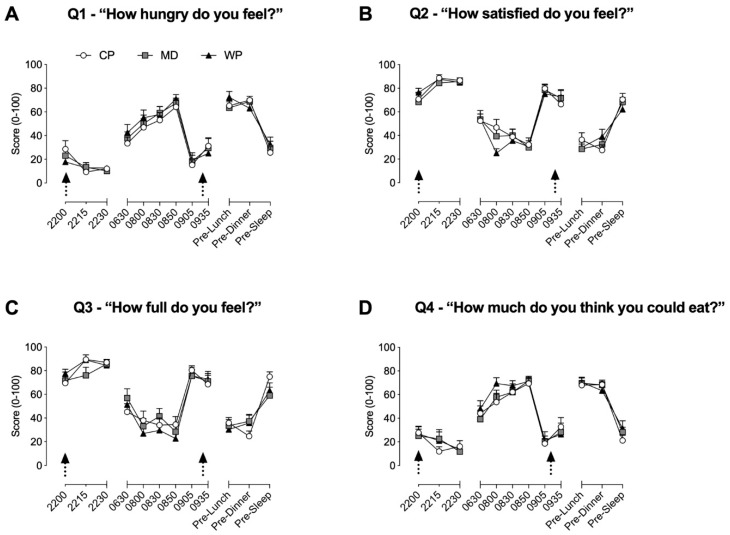
Subjective ratings of hunger (**A**), satisfaction (**B**), fullness (**C**) and desire to eat (**D**) obtained via the 100 mm visual analogue scale. Arrows represent the consumption of the pre-sleep treatment beverage and ad libitum breakfast, respectively. Data are presented as means ± SEM. No between-group significance was apparent, within group differences are stated in the results Section 3.1.

**Figure 3 nutrients-12-00090-f003:**
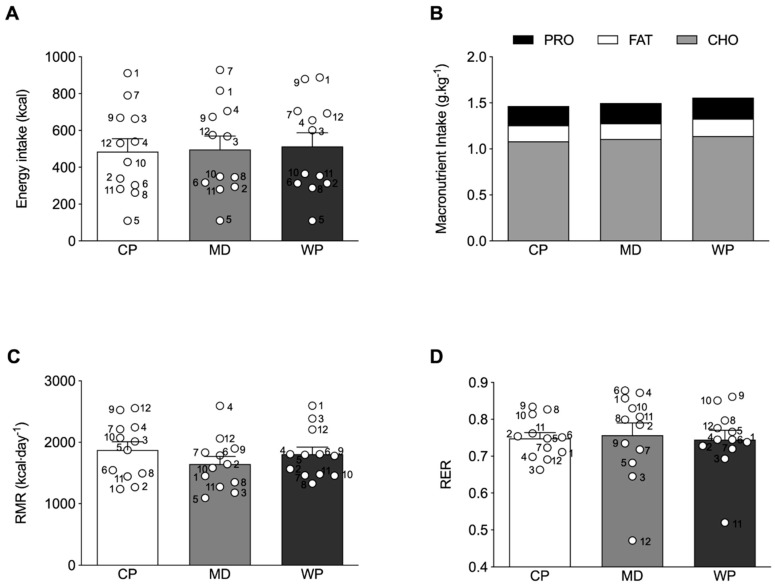
(**A**) Absolute total energy intake and (**B**) relative macronutrient intake at the ad libitum breakfast. (**C**) resting metabolic rate (RMR) and (**D**) respiratory exchange ratio (RER) as measured by indirect calorimetry. ○ represent individual values within each experimental trial, subject numbers are placed adjacent to the corresponding circle. Figures represent mean values with SEM.

**Figure 4 nutrients-12-00090-f004:**
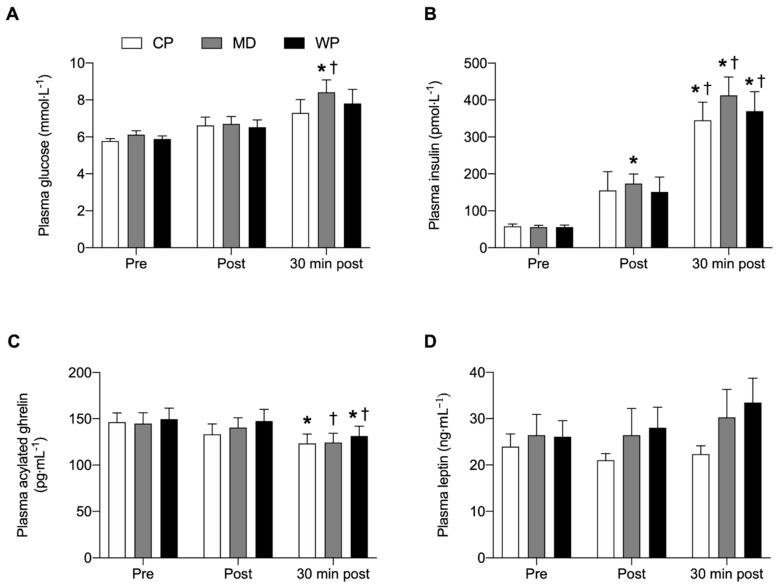
(**A**) Plasma glucose, (**B**) plasma insulin, (**C**) plasma acylated ghrelin and (**D**) plasma leptin concentrations immediately pre, post and 30 min post ad libitum breakfast consumption. Significance was set at *p* < 0.05. * Indicates significantly different from pre, ^†^ indicates significantly different from post. Values are represented as mean with SEM.

**Table 1 nutrients-12-00090-t001:** Nutritional content of bedtime beverages. (CP: Casein Protein, MD: Maltodextrin, WP: Water Placebo).

	Energy (kcal)	Carbohydrate (g)	Protein (g)	Fat (g)
CP	168	1.92	40	0.72
MD	168	42	0	0
WP	0	0	0	0

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
