# Peer review of "Pre-Sleep Casein Protein Ingestion Does Not Impact Next-Day Appetite, Energy Intake and Metabolism in Older Individuals"

_nutrients, 2019, doi:10.3390/nu12010090_

Round 1

Reviewer 1 Report

Night-time consumption of casein protein has been shown to enhance de novo muscle protein synthesis following resistance exercise, increase muscle protein synthesis, provide a more favourable whole-body protein balance overnight compared with fasting. This concept is relevant to older individuals as the current night-time feeding research has been conducted in elderly men, obese men and women, female runners and fit, college-aged men. The authors should place more emphasis on the exact importance of their study.

How much protein a day do people who participate in the study normally eat?

The authors must correct the bibliography, as there are typographical errors.

There are articles that I consider important and should appear cited as this topic is similar to the one they are dealing with and yet they are not in the text

Samantha M. Leyh, Brandon D. Willingham, Daniel A. Baur, Lynn B. Panton and Michael J. Ormsbee. Pre-sleep protein in casein supplement or whole-food form has no impact on resting energy expenditure or hunger in women. British Journal of Nutrition (2018), 120, 988–994

Jorn Trommelen and Luc J. C. van Loon. Pre-Sleep Protein Ingestion to Improve the Skeletal Muscle Adaptive Response to Exercise Training. Nutrients 2016, 8, 763; doi:10.3390/nu8120763

Tim Snijders, Jorn Trommelen, Imre W. K. Kouw, Andrew M. Holwerda, Lex B. Verdijk and Luc J. C. van Loon. The Impact of Pre-sleep Protein Ingestion on the Skeletal Muscle Adaptive Response to Exercise in Humans: An Update. 2019, Frontiers in Nutrition,  Volume 6, Article 17

Author Response

We thank the reviewer for their time to review our manuscript. We have amended the manuscript according to the reviewer's suggestions. Dietary data has been added as a supplemental file (supplemental file 2) and the suggested references have been added.

Reviewer 2 Report

My suggestion:

1.Discussion and Conclusion: authors should discuss assessment of long-term  consumption of pre-sleep protein intake on the muscle mass in the view of possible protection from sarcopenia. 

Author Response

We thank the reviewer for their time to review our manuscript. We have incorporated the suggestions made by the reviewer and hope this is to their satisfaction. We have added a brief statement in the conclusions reflecting the reviewer's comment.

Reviewer 3 Report

Dear authors,

Very interesting study due to the direct application in elderly nutrition.

However, please consider the following:

Line 17: Use "years" instead of "yrs".

Line 50: Authors should highlight the lack of side effects (i.e., kidney) during this protein intake. There are still misinformation sources and unbased claims like this.

Line 81: Do not start a sentence with digit numbers. Use "Twelve (12) healthy older participants..."

Line 87: Standardize ad libitum use (in line 77 authors use ad libitum but in line 87 use ad-libitum). Italics is recommended.

Line 95: Authors should report randomization method used in the study (i.e., minimization).

Line 98: Do you have total average sleep time? Sleep quality data missing. At least upload supplementary file.

Line 105: Figure 1 should show (or please remark) the pre-sleep treatment time.

Line 180: Do not start a sentence with digit numbers. Use "Ten mililiters (10 mL)..."

Line 184: Please specify manufacturer.

Lines 236: From my perspective, it would be interesting to report individual behavior of the data in figures 3. Please, mark each square/triangle/circle with the corresponding subject number.

Line 238: Please, standardise with either "standard error of mean" or the abbreviation "SEM" in all legend figures.

Line 253: WC or WP in figure 4? Please correct.

Line 270: Replace "triglycerides" by "triacylglycerols".

Lines 310-323: It would be interesting to see the behaviour of the sleep quality data (given the analysis that author did in this section). Even when there was non-significant changes, readers and other research groups may use the single-blinded cross-over data to base their future hypothesis/projects.

Line 358: Please introduce reference of Ormsbee et al. Also, it is important to highlight that this study was in female only and gave to the participants 12 g PRO and 30 g CHO.

Line 359: It is very important to correct what the cited researchers did. The authors state in line 359 "Ormsbee and colleagues found an increase in next-morning RMR when 335 mL of chocolate milk was consumed 7 h before bedtime"; however, the cited investigators did not use this methodology (FYI: "ingested chocolate milk ~30 min before sleep and 7-9 hrs before a morning exercise
trial" and "Subjects returned to the laboratory the following morning fasted (approximately 7-9 hrs after treatment consumption". Please, make the necessary changes.

Lines 360-362: Please mention the populations of the cited studies. They are different than the current investigation.

Lines 371-374: This is incorrect. Citation number 15 (Madzima et al., 2014) did not compare the effects of milk ingestion with whey protein or carbohydrates. In fact, the experimental groups of the cited study were: CHO, carbohydrate (maltodextrin); WP, whey protein; CP, casein protein; PLA, placebo (Propel Zero). Please make the necessary corrections.

Author Response

We thank the reviewer to have taken their time to review the current manuscript. We have changed the manuscript according to the reviewer's suggestions and feel this has improved the manuscript. We hope this is to the reviewer's satisfaction. Please see below for a point to point answer to the reviewer's comments:

Line 17: Use "years" instead of "yrs".

This has been changed to years

Line 50: Authors should highlight the lack of side effects (i.e., kidney) during this protein intake. There are still misinformation sources and unbased claims like this.

We fully agree with the reviewer and have added a brief statement indicating there are no negative consequences of increasing protein intake on health outcomes such as kidney function.

Line 81: Do not start a sentence with digit numbers. Use "Twelve (12) healthy older participants..."

This has been changed to twelve

Line 87: Standardize ad libitum use (in line 77 authors use ad libitum but in line 87 use ad-libitum). Italics is recommended.

We thank the reviewer for raising this point. We have gone through the manuscript and changed the wording to the following ad libitum

Line 95: Authors should report randomization method used in the study (i.e., minimization).

This information has been added (line 97).

Line 98: Do you have total average sleep time? Sleep quality data missing. At least upload supplementary file.

Unfortunately we did not measure average sleep time. We assessed sleep quality using the LSEQ. Participants’ scores for each question have been uploaded as a supplemental file (supplemental file 1).

Line 105: Figure 1 should show (or please remark) the pre-sleep treatment time.

We thank the reviewer for pointing this out. We have added the pre-sleep treatment in the figure.

Line 180: Do not start a sentence with digit numbers. Use "Ten mililiters (10 mL)..."

This has been changed to Ten

Line 184: Please specify manufacturer.

We apologise for omitting the information and have added the manufacturer’s details into the manuscript.

Lines 236: From my perspective, it would be interesting to report individual behavior of the data in figures 3. Please, mark each square/triangle/circle with the corresponding subject number. I am unsure how to make this graph.

We agree with the reviewer and have amended the figure. The corresponding subject numbers have been added to each symbol.

Line 238: Please, standardise with either "standard error of mean" or the abbreviation "SEM" in all legend figures.

This has been standardised to SEM in each figure legend

Line 253: WC or WP in figure 4? Please correct

This has been changed to WP.

Line 270: Replace "triglycerides" by "triacylglycerols".

This has been changed to triacylglycerols.

Lines 310-323: It would be interesting to see the behaviour of the sleep quality data (given the analysis that author did in this section). Even when there was non-significant changes, readers and other research groups may use the single-blinded cross-over data to base their future hypothesis/projects.

We fully agree with the reviewer that this information could be interesting for readers. We have therefore uploaded the sleep quality data as a supplemental file.

Line 358: Please introduce reference of Ormsbee et al. Also, it is important to highlight that this study was in female only and gave to the participants 12 g PRO and 30 g CHO.

This information has been added.

Line 359: It is very important to correct what the cited researchers did. The authors state in line 359 "Ormsbee and colleagues found an increase in next-morning RMR when 335 mL of chocolate milk was consumed 7 h before bedtime"; however, the cited investigators did not use this methodology (FYI: "ingested chocolate milk ~30 min before sleep and 7-9 hrs before a morning exercise
trial" and "Subjects returned to the laboratory the following morning fasted (approximately 7-9 hrs after treatment consumption". Please, make the necessary changes.

We thank the reviewer for their comment. We have changed the sentence to accurately reflect what Ormsbee and colleagues did.

Lines 360-362: Please mention the populations of the cited studies. They are different than the current investigation.

Study population specifics have been added to more accurately discuss present study findings.

Lines 371-374: This is incorrect. Citation number 15 (Madzima et al., 2014) did not compare the effects of milk ingestion with whey protein or carbohydrates. In fact, the experimental groups of the cited study were: CHO, carbohydrate (maltodextrin); WP, whey protein; CP, casein protein; PLA, placebo (Propel Zero). Please make the necessary corrections.

This section has been rewritten to more accurately reflect the study conducted by Madzima and colleagues.